# Prospects for Prostate Cancer Chemotherapy: Cytotoxic Evaluation and Mechanistic Insights of Quinolinequinones with ADME/PK Profile

**DOI:** 10.3390/biomedicines12061241

**Published:** 2024-06-03

**Authors:** Ayse Tarbin Jannuzzi, Ayse Mine Yilmaz Goler, Abanish Biswas, Subodh Mondal, Vinay N. Basavanakatti, Hatice Yıldırım, Mahmut Yıldız, Nilüfer Bayrak, Venkatesan Jayaprakash, Amaç Fatih TuYuN

**Affiliations:** 1Department of Pharmaceutical Toxicology, Faculty of Pharmacy, İstanbul University, 34116 İstanbul, Türkiye; tarbin.cevik@istanbul.edu.tr; 2Department of Biochemistry, School of Medicine/Genetic and Metabolic Diseases Research and Investigation Center, Marmara University, 34854 İstanbul, Türkiye; aysemine.yilmaz@gmail.com; 3Department of Pharmaceutical Sciences & Technology, Birla Institute of Technology, Mesra, Ranchi 835215, Jharkhand, India; abanish37@gmail.com (A.B.); drvenkatesanj@gmail.com (V.J.); 4Bioanalysis, Eurofins Advinus BioPharma Services India Pvt Ltd., Bengaluru 560058, Karnataka, India; subodh.mondal@advinus.com; 5Adgyl Lifesciences Private Limited, Bengaluru 560058, Karnataka, India; vinay.b@advinus.com; 6Department of Chemistry, Engineering Faculty, Istanbul University-Cerrahpasa, Avcılar, 34320 İstanbul, Türkiye; hyildirim@iuc.edu.tr; 7Department of Chemistry, Gebze Technical University, Gebze, 41400 Kocaeli, Türkiye; yildizm@gtu.edu.tr; 8Department of Chemistry, Faculty of Science, Istanbul University, Fatih, 34134 İstanbul, Türkiye; nbayrak@istanbul.edu.tr

**Keywords:** prostate cancer, reactive oxygen species, cytotoxicity, ADME, molecular dynamics

## Abstract

The evaluation of in vitro biological activity of several previously reported quinolinequinones (**AQQ1–5**) against 60 human cancer cell lines (NCI-60) used by the National Cancer Institute’s Developmental Therapeutics Program (DTP) contributed to our earlier research on possible anticancer and/or antibacterial agents. Of interest, NCI-60 screening revealed that two quinolinequinones (**AQQ1** and **AQQ2**) significantly reduced the proliferation of several cancer genotypes. Following the administration of a single dose and five additional doses, all quinolinequinones demonstrated a significant inhibitory effect on the growth of leukemia and other cancer cell lines. Hence, a series of subsequent in vitro biological assessments were performed to further understand the mechanistic impact of the compounds. In MTT assays, it was found that **AQQ1** and **AQQ2** exhibited higher efficacy against DU-145 cells (IC_50_ 4.18 µM and 4.17 µM, respectively) compared to MDA-MB-231 (IC_50_ 8.27 and 13.33 µM, respectively) and HCT-116 cells (IC_50_ 5.83 and 9.18 µM, respectively). Additionally, **AQQ1** demonstrated greater activity in this context. Further investigations revealed that **AQQ1** inhibited DU-145 cell growth and migration dose-dependently. Remarkably, arrest of the DU-145 cell cycle at G0/G1 phase and ROS elevation were observed. Pharmacokinetic (PK) studies revealed that **AQQ1** has better PK parameters than **AQQ2** with %F of 9.83 in rat. Considering the data obtained with human liver microsomal stability studies, **AQQ1** should have a better PK profile in human subjects. In silico studies (molecular dynamics) with three kinases (CDK2, CDK4, and MAPK) leading to cell cycle arrest at G_0_/G_1_ identified MAPK as a probable target for **AQQ1**. Taken together, our results showed that **AQQ1** could be a potential chemotherapeutic lead molecule for prostate cancer.

## 1. Introduction

The type of cancer caused by polyps, which is used to define any type of abnormal growth observed in the colon or rectum, is called colon cancer [1]. Although it is known that red meat consumption, alcohol, low folate levels, inappropriate lifestyle such as smoking, physical inactivity, or environmental factors increase the risk of disease, it has been determined that genetic factors play the leading role [2]. Germline mutations in some colon cancer genes are the most common known cause of hereditary colon cancer syndromes, occurring in 3–7% of all cases each year [3]. Also, it has been determined that the risk of colon cancer is high in first-degree relatives of people with colon cancer or colon adenomas that developed before the age of 60 [3,4].

Breast cancer is a type of cancer that usually does not show symptoms in its early stages, has minimal potential to spread, and is observed in the ducts or lobules arising from the lining cells (epithelium) of the ducts or lobules in the glandular tissue of the breast. However, it may progress from this place and spread to the surrounding breast tissue, lymph nodes, or other organs in the body over time [5]. Breast cancer can occur in women at any age after puberty, and the risk increases with age. Certain factors such as obesity, alcohol and tobacco use, genetic factors, and radiation exposure increase the risk of breast cancer [5,6]. Even if these risk factors can be controlled, the risk of developing breast cancer can only be reduced by 30% [5].

The prostate is one of the most important accessory glands of the male reproductive system [7]. Almost all prostate cancer, which occurs when the cells in the prostate gland begin to grow out of control, are adenocarcinomas, a type of cancer that originates from the mucus-secreting cells of the body called glandular cells [8]. There are also rare types of prostate cancer such as small cell carcinomas, neuroendocrine tumors, transitional cell carcinomas, and sarcomas. When a patient is diagnosed with prostate cancer, it can be said to be an adenocarcinoma. Although there are prostate cancers that can grow and spread rapidly, most of them are known to grow slowly. The autopsy result revealed that many elderly men with different causes of death had prostate cancer, which did not affect them at all throughout their lives, and they were unaware of its existence [9].

Since breast cancer targets women, prostate cancer targets males, and colon cancer threatens both sexes, causing millions of deaths worldwide each year, organic chemists and pharmaceutical chemists have joined forces to design new lead molecules that are target-selective, less toxic, and effective. The two most important results that emerged from the structure-activity studies carried out in order to better analyze the source of the biological activity of some natural products such as streptonigrin and LY83583 [10]: (a) the quinolinequinone moiety in these compounds is the structural motif responsible for the biological potential of the compounds and (b) the main reason why the quinolinquinone moiety is associated with biological activity is the redox cycle of these compounds and the accompanying with the reactive oxygen species (ROS) overproduction. These findings have made it a research topic for many scientists to obtain new biologically active synthetic molecules with a quinolinequinone moiety, which have a high tendency to undergo redox reactions. Scientific studies conducted by our group in this direction support that molecules with this moiety are building blocks that can be effective in both antimicrobial and cancer studies [11,12,13]. In light of the encouraging discoveries that have been made by our group, we engaged in comprehensive investigations aimed at identifying novel lead molecules for the treatment of cancer. These investigations build upon our previous work on the identification of new lead structures. In accordance with the Drug Evaluation Branch’s protocol, five quinolinequinones containing alkoxy groups were created and submitted to the NCI of Bethesda (Rockville, MD, USA) within the DTP [14,15] for testing on cell lines from nine distinct cancer types [16]. After single-dose testing, the DTP division of the NCI selected four quinolinequinones (**AQQ1–3** and **AQQ5**) for a full panel five-dose in vitro assay to ascertain their GI_50_ on the 60 cell lines. **AQQ1** and **AQQ2** were tested for cytotoxicity against the HCT-116 colon cancer, DU-145 prostate cancer, and MDA-MB-231/ATCC breast cancer cell lines in response to the encouraging NCI results. We also decided to find out the effects of the selected quinolinequinone (**AQQ1**) on prostate cancer cell proliferation, apoptosis, cell cycle, and the production of ROS. The host–guest interactions of potent quinolinequinones were also studied in detail using thorough in silico docking simulations.

## 2. Materials and Methods

### 2.1. Biological Evaluation

#### 2.1.1. In Vitro Single-Dose Anticancer Screening by NCI

The obtained quinolinequinones were submitted to NCI, Bethesda, USA, and, as per standard protocol of NCI, all compounds were evaluated for their antiproliferative activity at single-dose assay (10 µM concentration in DMSO) on a panel of 60 cancer cell lines derived from leukemia, non-small-cell lung, colon, CNS, melanoma, ovarian, renal, prostate, and breast, as per protocol. Tested compounds were added to the microtiter culture plates followed by incubation for 48 h at 37 °C. Sulforhodamine B (SRB), a protein-binding dye, is used for endpoint determination. The percent of the growth of the treated cells was determined in comparison to the untreated control cells and the results of each tested compound were reported. Data from one-dose experiments pertain to the percentage growth at 10 μM [16,17,18].

#### 2.1.2. In Vitro Five-Dose Anticancer Screening by NCI

Serial 5 × 10-fold dilution from an initial DMSO stock solution was performed, prior to incubation at each individual concentration. The most promising quinolinequinones (**AQQ1**, **AQQ2**, **AQQ3**, and **AQQ5**) were then elevated by DTP-NCI for a higher testing level to determine three dose–response parameters (GI_50_, TGI, and LC_50_) for each cell line after establishing a dose–response curve from 5 different concentrations 0.01, 0.1, 1, 10, and 100 µM for **AQQ1**, **AQQ2**, **AQQ3**, and **AQQ5**. The exact detailed procedure for the latter assay had been elaborated earlier.

#### 2.1.3. MTT Assay

For the MTT assay, DU-145, MDA-MB-231, HCT-116, and HUVEC cells (American Type Culture Collection) were seeded at a concentration of 1 × 10^4^ cells per well in 96-well plates and allowed to incubate overnight, ensuring proper adherence. Subsequently, a series of concentrations for both **AQQ1** and **AQQ2** (ranging from 1 to 100 µM) were added to the culture medium with a final concentration of 1% DMSO, and cells were subjected to an additional 24 h incubation period. The control group only received 1% DMSO (Biomatik, Cambridge, ON, Canada). To establish a baseline for comparison, DOXO (Saba, İstanbul, Turkey), a known chemotherapeutic agent, was administered at equivalent concentrations. Following the incubation period, MTT (3-(4,5-dimethylthiazol-2-yl)-2,5-diphenyltetrazolium bromide) (Biomatik, Canada) assay was carried out to assess cell viability [19]. Finally, the absorbance was quantified at 590 nm using a BioTek Epoc plate reader (Biotek, Winooski, VT, USA). The derived data were analysed using GraphPad Prism 8 software to calculate the inhibitory concentration 50 (IC_50_) values of the test compounds.

#### 2.1.4. Colony Formation Assay

DU-145 cells were seeded in 6-well plates at a density of 1000 cells/well and allowed to incubate overnight. Then, the cells were treated with varying concentrations of **AQQ1** (1, 2.5, and 5 µM) along with negative control and 2.5 µM DOXO for 24 h. After a medium refreshment, the cells were allowed to grow for 10 days. After incubation, the colonies are fixed with cold methanol for 5 min and then stained with a 0.5% crystal violet solution (prepared in 10% methanol) for 20 min. The plates were allowed to air-dry and the colonies were counted manually.

#### 2.1.5. Scratch Assay

DU-145 cells were grown in 6-well plates until they reached confluence. A straight-line scratch was then carefully created in the cell layer using a sterile pipette tip. The medium was replaced to remove any detached cells. The cells were treated with varying concentrations of **AQQ1** (1, 2.5, and 5 µM) along with negative control for 24 h and the cells were allowed to migrate into the denuded area. The brightfield images were taken under the microscope at 0 h and over 24 h. The extent of wound closure is quantified by measuring the change in the width of the scratch with Fiji Image J wound healing size macro [20].

#### 2.1.6. Apoptosis Necrosis Analysis

To assess cellular apoptosis and necrosis, a fluorochrome-labelled Annexin V (FITC) and Propidium Iodide (PI) staining kit (SONY Biotechnology, San Jose, CA, USA) was used according to the manufacturer’s protocol. DU-145 cells were initially seeded at a density of 1 × 10^6^ cells per well in 6-well plates. After an incubation period, the cells were treated with varying concentrations of **AQQ1** (1, 2.5, and 5 µM) along with negative control for 24 h. Then, the cells were detached using trypsin and then suspended in 500 µL of 1× Binding Buffer. Subsequently, 3 µL of Propidium Iodide (PI) and 5 µL of Annexin V-FITC were added to the cell suspension, followed by a 15 min incubation at room temperature in the dark. Post-incubation, cell analysis was performed via flow cytometry using the BD FACS Calibur Flow Cytometry System. The percentages of apoptotic and necrotic cells were determined using BD CellQuest Pro Software version 5.1 (both from Becton Dickinson, Sunnyvale, CA, USA).

#### 2.1.7. Cell Cycle Assay

Cells were subjected to **AQQ1** and DOXO treatments as previously described for apoptosis necrosis analysis. For cell cycle analysis, a cell population of 1 × 10^6^ was collected, centrifuged, and the supernatant was removed. The resulting cellular pellet was re-suspended in 70% cold ethanol in PBS. Subsequently, the cells underwent a single PBS wash, followed by incubation in PBS containing 50 mg/mL PI (Sigma-Aldrich, St. Louis, MO, USA) and 2 mg/mL DNase-free RNase-A (Affymetrix, Inc., Santa Barbara, CA, USA) for 30 min at room temperature in a light-protected environment. Analysis was performed using the BD FACS Calibur Flow Cytometry System, and the BD CellQuest Pro Software software (version 5.1, Becton Dickinson, USA) was employed to determine cell cycle phase distributions in comparison to the control group.

#### 2.1.8. Oxidative Stress

DU-145 cells were initially seeded at a density of 1 × 10^6^ cells per well in 6-well plates. Cells were subjected to **AQQ1** treatments as described earlier. After an incubation period, the cells were treated with varying concentrations of **AQQ1** (1, 2.5, and 5 µM) for 24 h. ROS analysis was carried out using the DCFDA oxidation method according to Eruslanov and Kusmartsev with some modifications [21]. For positive control, the cells were exposed to 100 µM H_2_O_2_ for 30 min. Then, the medium was discarded, 20 µM DCDFA (Sigma-Aldrich, USA) in culture medium was added to the wells, and plates were incubated for 15 min in the dark. The cells were analysed with NovoCyte (Agilent, Palo Alto, CA, USA) flow cytometry using NovoExpress version 1.6.2 software (Agilent, USA).

#### 2.1.9. Statistical Analysis

For cell culture studies, the data analysis was conducted using GraphPad 8 software (USA). Each experiment was carried out at minimum in triplicate, and the results are presented as the mean ± standard error of the mean (SEM). Shapiro–Wilk test was performed to verify the normality of the data. Hence, group means were compared with control means using a one-way ANOVA with Dunnett’s multiple comparison test. A *p* value less than 0.05 was considered statistically significant.

### 2.2. ADME and PK Profiling

#### 2.2.1. In Vitro Metabolic Stability Study

For experimental details, please see the Appendix A.

#### 2.2.2. In Vivo Bioavailability Study of AQQ1 and AQQ2 in Male Sprague-Dawley Rats

For experimental details, please see the Appendix A.

#### 2.2.3. Bioanalytical Method (LC/MS/MS) for AQQ1 and AQQ2

For experimental details, please see the Appendix A.

### 2.3. Molecular Docking and Molecular Dynamics

Molecular docking and dynamics simulations were carried out as per the protocol elaborated in our earlier publications [22,23,24]. The experimental XRD 3D structure of selected target proteins was retrieved from the Protein Data Bank (PDB). Missing residues were constructed, and proteins were prepared for docking using the Modeler and Dockprep module implemented in UCSF Chimera [25]. Autodock parameter files (protein.gpf and ligand.dpf) were prepared using Python utility scripts from MGLTools-1.5.7. Docking simulation was carried out in Google Colab using a GPU-enabled virtual machine. AutoDock-GPU [26] was used to run the molecular docking simulation with 300 runs for each ligand. For interaction analysis, the lowest energy conformer from the largest cluster was considered. Python utility scripts from MGLTools-1.5.7 were used to pick the lowest energy conformer from the largest cluster and to prepare the complex.pdb files. These complex.pdb files were then subjected to molecular dynamics simulation using GROMACS 2019 package. CHARMM27 all-atom force field was used and the topology file for ligands was generated through the SwissParam (https://www.swissparam.ch/) webserver. A system having a solvated complex (in a cubic box of TIP3P water molecules) neutralized with the counter ions (Na^+^ and Cl^−^) was subjected to simulation using an NPT ensemble with periodic boundary conditions (300 K, 1 atm). The Leapfrog algorithm was used to integrate the equation of motion with a time (2 fs), and the long-range electrostatic interactions with a cut-off distance of 12 Å were calculated using the particle mesh Ewald method. The simulations were performed for a duration of 300 ns, and the trajectory was captured with a 10 ps gap. The RMSD, RMSF, H-bond, and ROG were analysed using Xmgrace (https://plasma-gate.weizmann.ac.il/Grace) or R-package (version 4.1.2). LigPlot+ [27] was used to capture the 2D interaction plot of complex coordinates extracted from MD trajectories.

## 3. Results and Discussion

### 3.1. Biological Evaluation

#### 3.1.1. Preliminary Anticancer Screening

NCI has preliminary assessed the in vitro anticancer activity of alkoxy-substituted aminoquinolinequinones (**AQQ1–5**), presented in Figure 1 and previously produced according to the literature, against cancer cell lines. **AQQ1–5** were synthesized according to the literature [28]. The growth percentage (GP) and lethality (values less than 0) of the treated cells were listed using the collected data in Table 1. Whereas **AQQ1** displayed moderate potency against non-small cell lung cancer EKVX and NCI-H226 cell lines (71.48% and 70.35% inhibition, respectively) and prostate cancer PC-3 cell line (with 72.61% inhibition), it showed superior biological activity against SF-539 (98.22% inhibition) of CNS cancer cell lines, SK-MEL-28 (85.87% inhibition) of melanoma cell lines, and both OVCAR-8 and NCI/ADR-RES (90.13% and 92.92% inhibition, respectively) of ovarian cancer cell lines. It was determined that **AQQ2** showed maximum sensitivity against OVCAR-8 with 99.07% inhibition, HS 578T with 98.95% inhibition, and K-562 with 98.74% inhibition of tumor cell growth. **AQQ2** also showed promising anticancer activity against the NCI-H226 with 82.38% inhibition, NCI-H322M with 71.16% inhibition, and NCI-H460 with 87.93% inhibition of non-small cell lung cancer cell lines, HCC-2998 with 79.65% inhibition and HT29 with 80.04% inhibition of colon cancer cell lines, SK-MEL-28 with 85.45% inhibition of melanoma cell line, and PC-3 with 95.66% inhibition of breast cancer cell line. The **AQQ3** turned out to be a lead molecule with remarkable values specifically against CCRF-CEM cell line (99.37% inhibition), NCI-H460 cell line (86.59% inhibition), HT29 cell line (84.10% inhibition), U251 cell line (84.40% inhibition), SK-MEL-28 cell line (91.07% inhibition), OVCAR-8 cell line (98.97% inhibition), 786-0 cell line (91.61% inhibition), and PC-3 cell line (89.60% inhibition). Interestingly, the **AQQ4** displayed promising anticancer activity against the CCRF-CEM with 84.71% inhibition, HL-60(TB) with 90.05% inhibition, K-562 with 92.82% inhibition, and SR with 99.08% inhibition. Additionally, two types of ovarian cancer (IGROV1 and OVCAR-3) with 80.37 and 86.84% inhibition and breast cancer cell line (T-47D) with 85.07% inhibition were determined to be susceptible to **AQQ4**. **AQQ5** showed biological activity against CCRF-CEM, K-562, MOLT-4 (97.28%, 95.27%, and 91.71% inhibition), HOP-62 and NCI-H460 (99.15% and 97.98% inhibition), HCC-2998 and HT29 (85.56% and 75.09% inhibition), SK-MEL-2 (94.97% inhibition), OVCAR-5 (78.00% inhibition), and PC-3 (72.01% inhibition).

#### 3.1.2. In Vitro Full Panel Five-Dose 60-Cell Lines Assay

**AQQ1**, **AQQ2**, **AQQ3**, and **AQQ5** were used to explore against a panel of 60 human cancer cell lines more deeply at a five-dose screening at five different concentrations (100, 10, 1.0, 0.1, and 0.01 μM) after satisfying the threshold inhibition criteria (GI_50_, TGI, and LC_50_) of NCI in the NCI single-dose assay. TGI indicates total growth inhibition, while GI_50_ represents the concentration at which 50% of the growth inhibitory effect occurs, and LC_50_ value designated the concentration at which 50% of cancer cells death [16,17] shown in Table 2 (all in µM). In vitro activity was considered for lead molecules (**AQQ1**, **AQQ2**, **AQQ3**, and **AQQ5**) that inhibited cell line proliferation by 30% or less (negative numbers imply kills).

The GI_50_ values presented in Table 2 reveal that most tested quinolinequinones had a GI_50_ of less than 5 μM against most cell lines. Four quinolinequinones showed remarkable anticancer activity against most of the tested cancer cell lines. Four quinolinequinones were found to be highly sensitive towards leukemia K-562 and SR cell lines (with GI_50_ = 0.29 μM to 1.17 μM and LC_50_ ≥ 100). Furthermore, they showed valuable inhibition towards most of the tested cancer cell lines with a GI_50_ below 3.00 µM. All tested quinolinequinones possessed promising cytotoxic activity with TGI values not more than 10 µM except for most of the leukemia cell lines and SK-OV-3 cell line of ovarian cancer. Aside from the HL-60(TB) cell line for **AQQ1** and **AQQ2**, all leukemia cell lines in the panel had LC_50_ values more than 100 µM. Regarding the lethality, they mostly displayed LC_50_ values below 30 µM against the mentioned cancer cell lines except most of the leukemia and some of the breast cancer cell lines. Figure 2 shows all of the **AQQ1**’s five-dose response curves against the whole panel of 60 human cancer cell lines.

#### 3.1.3. Cytotoxicity Study of **AQQ1** and **AQQ2** with MTT Assay

In this section, the effects of **AQQ1** and **AQQ2** were evaluated alongside Doxorubicin HCl (DOXO) (a well-established chemotherapeutic agent) across versatile cell lines: DU-145, MDA-MB-231, HCT-116, and Human Umbilical Vein Endothelial Cells (HUVEC). Comparing the cell lines, it is evident that they displayed varying sensitivities to the tested substances (Figure 3).

The IC_50_ values were used to assess cytotoxicity [29]. Comparing the cell lines, it is evident that they displayed varying sensitivities to **AQQ1** and **AQQ2**. Among the cancer cell lines, DU-145 exhibited the highest sensitivity to both **AQQ1** and **AQQ2** with IC_50_ values of 4.18 ± 0.73 µM and 4.17 ± 0.8 µM, respectively (Table 3). HCT-116 and MDA-MB-231 showed intermediate sensitivities to **AQQ1** and **AQQ2**, while HUVEC, a non-cancerous cell line, was less affected than DU-145 by these substances, which can indicate the selectiveness of the compounds. In contrast, DOXO, the known chemotherapeutic agent, displayed consistently higher IC_50_ values, indicating its lower cytotoxicity across all cell lines. DOXO exhibited the most significant cytotoxic effect in HCT-116 (24.35 ± 6.9 µM), followed by HUVEC (IC_50_ = 57.48 ± 9.82 µM), MDA-MB-231 (59.06 ± 9.06 µM), and had no impact DU-145 (IC_50_ < 100 µM) on tested concentrations. Taken together with NCI results, the observation indicates that the position of alkoxy group (ortho position) is associated with higher cytotoxic activity. Based on these results, while both **AQQ1** and **AQQ2** demonstrated cytotoxicity against the tested cancer cell lines, DU-145 was the most sensitive among them. **AQQ1** showed slightly higher cytotoxic activity in DU-145 cells and further investigations are carried out to explore the potential of **AQQ1** as cancer treatment option and to understand its mechanism of action better.

The colony formation assay is a valuable tool in cancer drug investigation, allowing researchers to assess the long-term effects of potential treatments on cancer cells by measuring their ability to form colonies, reflecting their growth and survival capabilities. As can be seen in Figure 4A, **AQQ1** significantly reduced the colony-forming ability of DU-145 cells. Complementing these findings with scratch assay results (which provide insights into the potential anti-metastatic properties of **AQQ1** by assessing its impact on cancer cell migration and invasion) we observed that DU-145 cell migration was inhibited by 13.37% ± 7.07 with 1 μM **AQQ1**, by 23.68% ± 7.39 with 2.5 μM **AQQ1**, and by 49.77% ± 8.45 with 5 μM **AQQ1** (Figure 4B). Furthermore, both our recent studies and those conducted by other research groups have reported on the cytotoxic and antiproliferative activities of various quinoline derivatives [12,30].

To evaluate the cell cycle-altering potential of **AQQ1**, flow cytometry analysis was employed. The results revealed that only the highest concentrations of **AQQ1** significantly increased the population of cells in the G0/G1 phase (Figure 5). Additionally, the positive control drug DOXO showed a similar effect on DU-145 cells. This indicates that **AQQ1** induces cell cycle arrest specifically at the G0/G1 phase in DU-145 cells, contributing to its notable anticancer activity against this cell line. The present observation is significant because halting the cell cycle at specific checkpoints can be an effective strategy in cancer therapy.

Further evaluation was carried out with an assessment of apoptosis and necrosis induction with **AQQ1** treatment in DU-145 cells. According to our results, **AQQ1** reduced live cell counts and increased necrotic cells which indicates dose-dependent modulation of cell death (Figure 6).

To further investigate the toxicity mechanisms involved in the anticancer activity of **AQQ1**, we conducted flow cytometric measurements of intracellular ROS levels using the 2′,7′-dichlorofluorescein-diacetate (DCDFA) probe after 24 h of **AQQ1** treatment in DU-145 cells. Figure 7 illustrates the results showing that **AQQ1** significantly increased ROS levels dose-dependently. These findings suggest that **AQQ1** treatment leads to increased ROS levels in DU-145 cells, indicating a potential mechanism of action for its anticancer activity, possibly involving the initiation of apoptosis.

Supporting our findings, previous studies demonstrated that different quinolinone hybrids exhibit high in vitro anticancer activity. For instance, quinolinone-benzimidazole and quinolinone-thiophene hybrids could exert antitumor effects through the modulation of cell cycle arrest, apoptosis, and ROS generation in A549 and HepG2 cells, respectively [31,32].

### 3.2. Pharmacokinetic Profiling

#### 3.2.1. In Vitro ADME Studies

Based on the anticancer profiles of the molecules, **AQQ1** and **AQQ2** were selected for in vitro ADME studies (Log P, Log D, and metabolic stability studies, shown in Table 4) and were found to be better than positive controls (Verapamil and Atenolol). Both the molecules did not violate the Lipinski’s rule of five (Ro5) and had the experimental Log P and Log D values of the 1.77–1.85 and 2.35–2.37, respectively. One can expect the compounds to have good oral absorption. Metabolic stability studies using liver microsomes of mouse, rat, dog, and human revealed that both compounds were having better half-life and low hepatic clearance compared with Verapamil in human liver microsomal fraction (For experimental details, please see the Appendix A).

#### 3.2.2. In Vivo Pharmacokinetic Studies

Using the validated Phoenix^®^ WinNonlin^®^ 8.3 NCA tool Version 8 or higher (Certara L.P.), USA, pharmacokinetic parameters were computed. As appropriate, C_max,_ T_max_, and exposures (AUC_infinity_ and AUC_last_) were calculated. Furthermore, pharmacokinetic characteristics such as distribution volume (V_d_), hepatic clearance (C_L_), elimination half-life (T_1/2_), and parenteral route like (C_0_) were evaluated. Using dose-normalized non-intravenous exposure versus dose-normalized intravenous exposure, oral bioavailability (F%) was computed (Table 5).

The mean plasma clearance of male Sprague–Dawley rats after a single IV bolus injection of **AQQ1** formulation (1 mg/kg) was found to be extremely high at 161 mL/min/kg, which was approximately 2.93 times greater than the rats’ typical hepatic blood flow. The mean volume of distribution was 90.3 L/kg, which was approximately 1290-folds greater than 0.7 L/kg total body fluids indicating, highly distributed in tissues. In male rats, the average terminal plasma half-life was 9.35 h. Male Sprague–Dawley rats were given a single peroral dosage formulation of **AQQ1** (5 mg/kg) and the median time to achieve peak plasma concentration was 0.5 h. The C_max_ was 435 ng/mL and their AUC_last_ (plasma exposure) was 3810 ng × h/mL. Calculated oral bioavailability was 9.83%.

Following a single IV bolus administration (1 mg/kg) of **AQQ2** formulation to male Sprague–Dawley rats, the mean plasma clearance was found to be very high, at 154 mL/min/kg. This value is approximately 2.8-fold higher than the normal hepatic blood flow of rats. The mean volume of distribution was 147 L/kg, which was approximately 210-fold greater than 0.7 L/kg total body fluids, indicating a highly distributed state in tissues. In male rats, the average terminal plasma half-life was 12.8 h. After giving male Sprague–Dawley rats a single oral dosage formulation of **AQQ2** (5 mg/kg), the median time to achieve peak plasma concentration was 1 h, with a C_max_ of 282 ng/mL and an AUClast (plasma exposure) of 2300 ng*h/mL. Calculated oral bioavailability was 5.96% (For experimental details, please see the Appendix A).

### 3.3. In Silico Molecular Interaction Analysis

Based on the anticancer profile and their effect on cell cycle, the following proteins were selected to identify the putative target: CDK2, CDK4, and MAPK. Any compound that impairs the function of these proteins is reported to inhibit the cell cycle at the G0/G1 phase [33,34]. Hence, we performed molecular docking simulation (AutoDock-4.2) for **AQQ1** and **AQQ2** with experimental X-ray crystallographic 3D structure of CDK2 (PDB: 6GUB), CDK4 (PDB: 7SJ3), and MAPK (PDB: 5UOJ) following the protocol reported earlier in our publications. Both analogs docked well into the active site pocket of all the three target proteins, but with poor scores in comparison with the co-crystallized ligand. We then subjected the complex to molecular dynamics (MD) simulation (GROMACS) to mimic conditions close to the biological environment and to study the interactions at an atomistic level. The MD simulation was carried out for a duration of 300 ns. Analysis of RMSD plot (protein backbone and ligand) of the six complexes revealed that only **AQQ1**-MAPK complex exhibited stability throughout 300 ns duration after initial 20 ns (Figure 8b). During the period of simulation, it was able to establish 2–4 H-bonding interactions (Figure 8c), but all with solvent molecules within the pocket. This clearly shows the interaction between **AQQ1** and MAPK is purely hydrophobic (Figure 8a). The fact that the pocket was not completely desolvated tells us that the interaction between **AQQ1** and MAPK is not so tight and **AQQ1** may be a moderate-to-weak binder of MAPK. On the other hand, **AQQ2** displayed the stability only for the first 50 ns (Figure 8e). Analysis of the snapshot beyond the first 50 ns revealed that it has been squeezed away from the binding pocket and is largely solvent exposed (Figure 8d), establishing H-bonds with solvent, GLN138, and CYS167 (Figure 8d,f). This clearly shows that replacing methoxy substitution with ethoxy makes the molecule a weak binder against MAPK. This simulation results suggests that MAPK may be one of the possible targets for **AQQ1**. A kinase panel screening may reveal other possible kinase targets of **AQQ1** and **AQQ2**.

## 4. Conclusions

In this study, we introduced the aminated quinolinequinones (**AQQ1–5**) containing alkoxy substituent(s) within the amino moiety that have previously been described by our group [28] as a new class for anticancer lead candidates. The aminated quinolinequinones (**AQQ1–5**) showed high biological potency against cancer cell lines. In order to evaluate the dose-response curves in the 60-cell line panel, the NCI forwarded four of them to the five-dose screening stage. For the purposes of assessing the activity of quinolinequinones, an agent with a GI_50_ value of less than 2 μM is considered powerful and may represent selectivity towards that specific cancer cell line. Consequently, four AQQs exhibited a highly anticancer profile, with low micromolar GI_50_ and TGI values against the majority of cancer cell lines. In response to the promising NCI results, two quinolinequinones (**AQQ1** and **AQQ2**) were subjected to an in vitro cytotoxicity evaluation against the HCT-116, DU-145, and MDA-MB-231 cell lines. **AQQ1** exhibits promising activity against DU-145 of prostate cancer cells. This effect involves the arrest of the cell cycle, the generation of ROS, and the induction of apoptosis, which highlights its potential in prostate cancer therapy and makes it a potential candidate for further development as a chemotherapeutic agent. On the other hand, our group will focus further on quinolinequinones’ ability to precisely target leukemia cancer cell lines. In vitro metabolic stability studies revealed quick metabolism and hepatic clearance with rat liver microsomes, but the in vivo PK studies revealed that absorption plays a limiting role and hence **AQQ1** was found to have mean plasma terminal half-life of 9.35 h. This means **AQQ1** may take around two days to reach a steady state, may require less frequent dosing, and has the potential to get accumulated (associated toxicity may be expected). Considering these factors, the dose required for eliciting required pharmacological effect and dosing frequency should be determined during subsequent drug discovery pipeline. Three kinases (CDK2, CDK4, and MAPK) that are known to cause cell cycle arrest in the G0/G1 phase were taken into consideration in order to determine the potential target for these drugs. These kinases were selected for simulation studies. While results of molecular docking against all the three targets were convincing for both **AQQ1** and **AQQ2**, molecular dynamics simulation proved that only MAPK as probable target with better interaction with **AQQ1**. A kinase profiling of **AQQ1** will enable us to substantiate the claim in the near future.

## Figures and Tables

**Figure 1 biomedicines-12-01241-f001:**
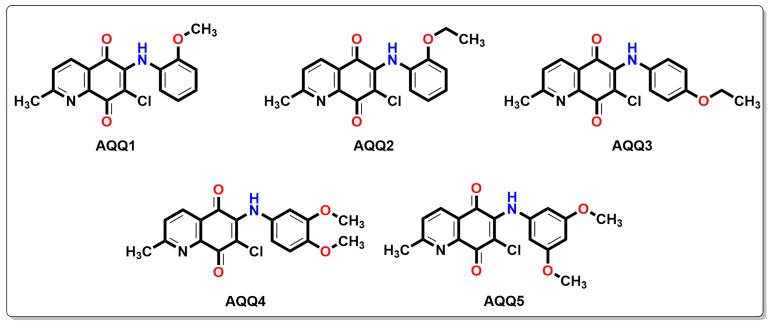
Chemical structures of alkoxy substituted aminoquinolinequinones (**AQQ1–5**) involved in current exploration.

**Figure 2 biomedicines-12-01241-f002:**
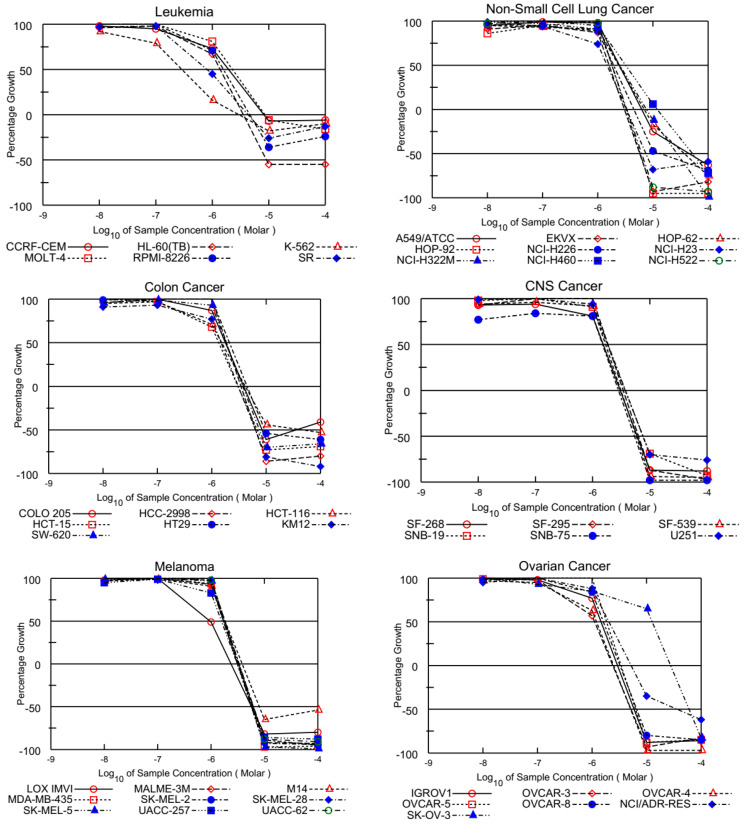
Growth inhibition of **AQQ1** at five-dose assay.

**Figure 3 biomedicines-12-01241-f003:**
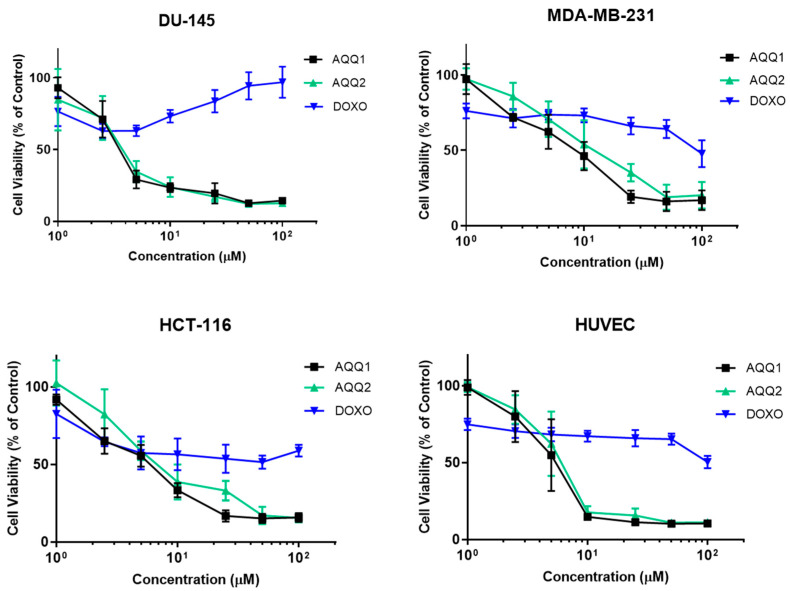
Cytotoxic effect of **AQQ1** and **AQQ2** on DU-145, MDA-MB-231, HCT-116, and HUVEC is evaluated by MTT assay after 24 h treatment. Values expressed as mean ± SEM, *n* = 6.

**Figure 4 biomedicines-12-01241-f004:**
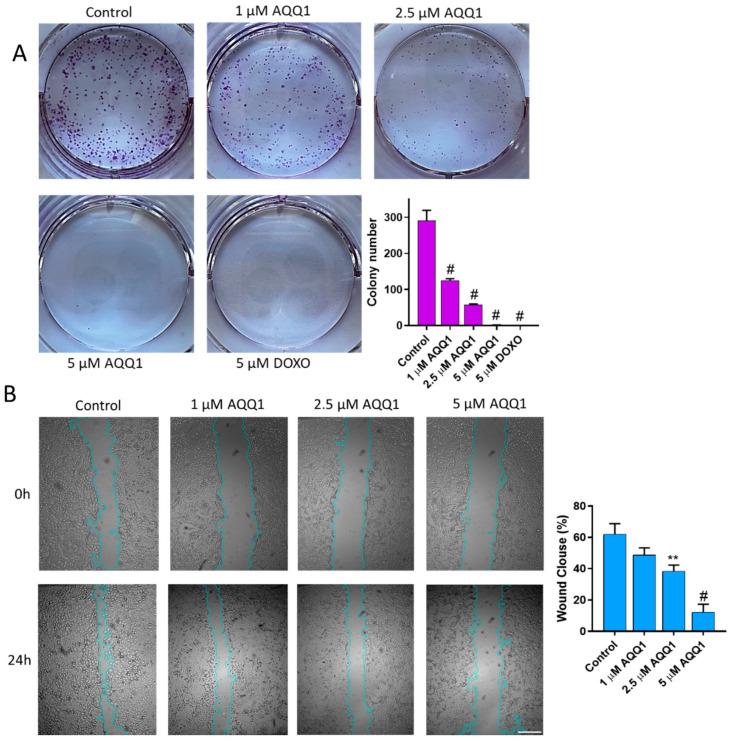
Representative images with quantitative analyses of colony formation (**A**) and scratch assays (**B**) in DU-145 cells following **AQQ1** treatment. ^#^ *p* < 0.0001, ** *p* < 0.01. Scale bar = 100 μm, *n* = 3.

**Figure 5 biomedicines-12-01241-f005:**
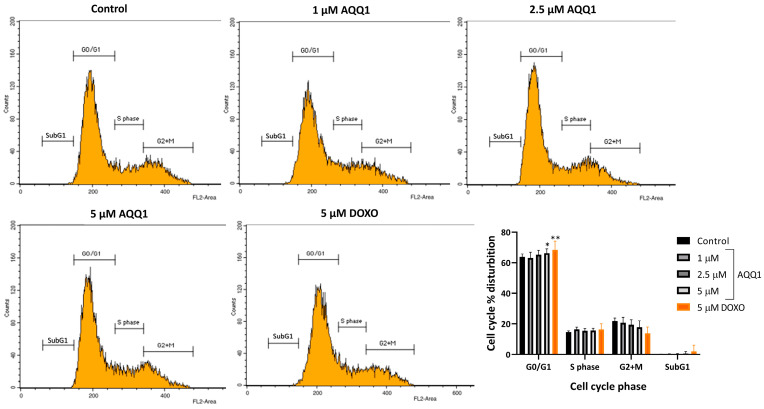
Representative images along with quantitative analysis of cell cycle distribution in DU-145 cells following **AQQ1** treatment. Values expressed as mean ± SEM. ** *p* < 0.01, * *p* < 0.05, *n* = 4.

**Figure 6 biomedicines-12-01241-f006:**
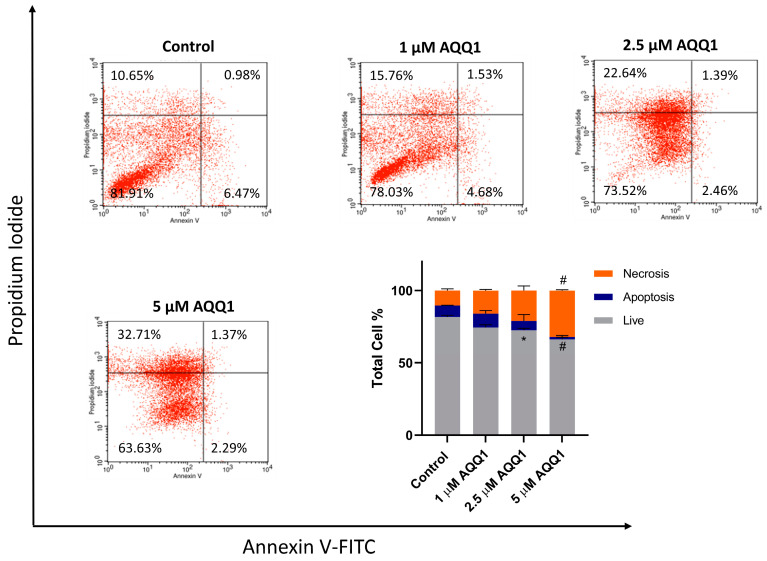
Representative images from flow cytometric apoptosis/necrosis assay, along with quantitative analysis for DU-145 cells treated with **AQQ1**. Values expressed as mean ± SEM. ^#^ *p* < 0.0001, * *p* < 0.05, *n* = 3.

**Figure 7 biomedicines-12-01241-f007:**
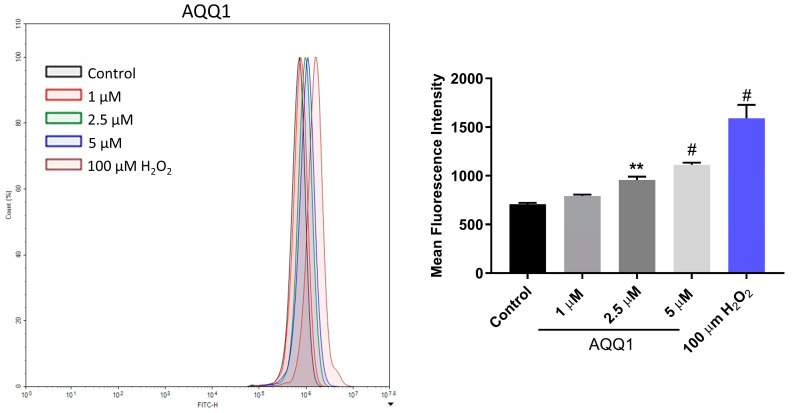
Representative images with quantitative analysis of ROS production in DU-145 cells following **AQQ1**. Values expressed as mean ± SEM. ^#^ *p* < 0.0001, ** *p* < 0.01, *n* = 3.

**Figure 8 biomedicines-12-01241-f008:**
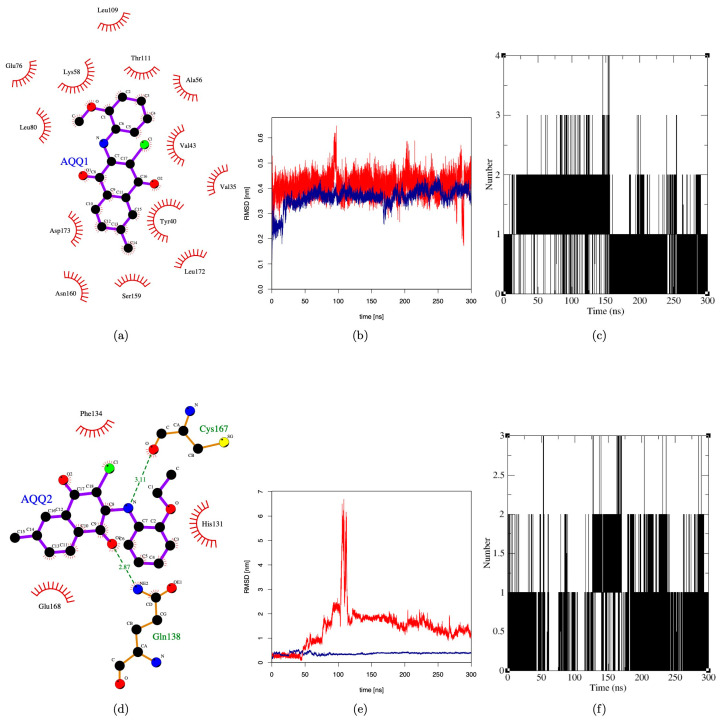
Molecular dynamics simulation of **AQQ1** and **AQQ2** against MAPK. (**a**,**d**) Two-dimensional interaction plot of **AQQ1** and **AQQ2** with MAPK; (**b**,**e**) RMSD plot of **AQQ1** and **AQQ2** for duration of 300 ns; and (**c**,**f**) H-bond plot of **AQQ1** and **AQQ2** for duration of 300 ns.

**Table 1 biomedicines-12-01241-t001:** Anticancer activity results as per single-dose assay at 10 µM concentration as GP of **AQQ1–5** from NCI 60 cell one-dose screen.

Cancer Cell Lines		AQQ1	AQQ2	AQQ3	AQQ4	AQQ5
**Leukemia**						
GP	CCRF-CEM	−4.94	−6.33	0.63	15.29	2.72
	HL-60(TB)	−48.05	−36.75	−46.39	9.95	−13.44
	K-562	−21.12	1.26	−4.17	7.18	4.73
	MOLT-4	−39.00	−24.89	−46.73	69.83	8.29
	RPMI-8226	−30.44	−35.12	−19.35	47.33	−39.66
	SR	−27.92	−12.75	−27.42	0.92	−3.19
**Non-Small Cell Lung Cancer**						
GP	A549/ATCC	101.36	59.11	46.18	107.06	42.41
	EKVX	28.52	−2.24	−97.48	50.81	74.66
	HOP-62	74.12	84.25	72.15	110.03	0.85
	HOP-92	−83.81	−89.25	−86.99	112.07	−12.23
	NCI-H226	29.65	17.62	−52.55	97.11	68.59
	NCI-H23	−8.37	−67.81	−61.00	59.03	−55.61
	NCI-H322M	71.14	28.84	32.37	99.05	89.75
	NCI-H460	70.83	12.07	13.41	100.46	2.02
	NCI-H522	−48.50	−43.48	−61.57	78.84	−80.61
**Colon Cancer**						
GP	COLO 205	111.34	99.24	98.22	107.70	−80.52
	HCC-2998	58.90	20.35	−55.70	103.69	14.44
	HCT-116	−80.64	−79.44	−83.64	55.84	−97.84
	HCT-15	−47.03	−76.56	−83.19	82.29	−66.21
	HT29	75.04	19.96	15.90	109.56	24.91
	KM12	69.40	60.77	57.40	98.23	39.30
	SW-620	−63.13	−66.65	−55.33	72.23	−90.16
**CNS Cancer**						
GP	SF-268	47.42	32.75	32.33	84.69	37.00
	SF-295	96.61	90.72	51.05	103.62	51.57
	SF-539	1.78	−37.33	−58.82	103.46	−30.92
	SNB-19	38.92	46.47	44.68	94.62	30.42
	SNB-75	105.38	101.38	96.54	110.92	−4.09
	U251	62.79	37.89	15.60	105.80	36.76
**Melanoma**						
GP	LOX IMVI	−72.99	−82.20	−76.37	−3.09	−82.62
	MALME-3M	−55.92	−71.39	−90.58	94.54	−89.37
	M14	ND ^a^	ND ^a^	ND ^a^	ND ^a^	−95.99
	MDA-MB-435	−85.96	−85.72	−83.74	−67.16	−87.43
	SK-MEL-2	78.82	−15.72	−49.77	100.13	5.03
	SK-MEL-28	14.13	14.55	8.93	101.65	−73.06
	SK-MEL-5	−93.34	−97.76	−98.64	86.57	−74.93
	UACC-257	−82.67	−88.33	−89.19	91.03	−92.05
	UACC-62	−45.84	−29.14	−43.31	71.83	−43.56
**Ovarian Cancer**						
GP	IGROV1	−83.51	−57.52	−82.43	19.63	−23.13
	OVCAR-3	CN ^b^	CN ^b^	CN ^b^	13.16	CN ^b^
	OVCAR-4	−98.22	−91.97	−97.52	71.38	−93.28
	OVCAR-5	78.91	40.78	54.83	104.66	22.00
	OVCAR-8	9.87	0.93	1.03	92.63	−0.82
	NCI/ADR-RES	7.08	−9.58	−15.63	94.66	−1.26
	SK-OV-3	88.34	85.34	79.78	91.82	93.82
**Renal Cancer**						
GP	786-0	9.17	−88.18	8.39	98.99	−59.47
	A498	71.81	78.84	100.96	82.55	71.10
	ACHN	−99.93	−99.15	−98.42	95.00	−99.62
	CAKI-1	ND ^a^	ND ^a^	ND ^a^	ND ^a^	−96.53
	RXF 393	ND ^a^	ND ^a^	ND ^a^	ND ^a^	−97.24
	SN12C	−49.46	−50.97	−57.33	90.46	−63.27
	TK-10	147.78	124.08	121.08	148.82	−50.07
	UO-31	−65.68	−92.96	−65.05	100.30	−94.65
**Prostate Cancer**						
GP	PC-3	27.39	4.34	10.40	70.56	27.99
	DU-145	−57.26	−85.02	−92.73	93.00	−92.99
**Breast Cancer**						
GP	MCF7	−60.38	−59.68	−66.97	59.00	−51.73
	MDA-MB-231/ATCC	−57.30	−63.63	−71.74	72.96	−28.56
	HS 578T	72.15	1.05	48.97	118.57	−32.78
	BT-549	83.54	44.40	46.48	126.22	24.11
	T-47D	−44.73	−40.00	−43.85	14.93	−56.01
	MDA-MB-468	−79.74	−90.72	−95.15	58.11	−78.49

^a^ ND means “not determined”; ^b^ cell lines are cancelled because cell line identity was not authenticated. Values between 0 and 100 indicate growth inhibition, values less than 0 indicate lethality.

**Table 2 biomedicines-12-01241-t002:** Threshold inhibition criteria values (in µM) as per five-dose assay of **AQQ1**, **AQQ2**, **AQQ3**, and **AQQ5** after 48 h based on SRB assay at NCI.

Molecule	**AQQ1 (NCI: D-829432/1)**			**AQQ2 (NCI: D-829433/1)**			**AQQ3 (NCI: D-829431/1)**			**AQQ5 (NCI: D-830803/1)**		
Panel/Cell Line	GI_50_	TGI	LC_50_	GI_50_	TGI	LC_50_	GI_50_	TGI	LC_50_	GI_50_	TGI	LC_50_
**Leukemia**												
CCRF-CEM	1.94	8.21	>100	1.41	>100	>100	2.11	>100	>100	2.23	ND ^a^	>100
HL-60(TB)	1.37	3.52	9.03	1.03	2.92	8.29	1.47	4.18	>100	1.38	4.65	>100
K-562	0.290	2.94	>100	0.341	ND ^a^	>100	1.17	>100	>100	0.381	>100	>100
MOLT-4	2.27	8.60	>100	2.03	8.09	>100	2.49	>100	>100	1.50	7.00	>100
RPMI-8226	1.57	4.61	>100	1.20	4.71	>100	1.67	5.33	>100	1.79	5.19	>100
SR	0.794	4.25	>100	0.543	4.14	>100	1.09	>100	>100	0.350	2.43	>100
**Non-Small Cell Lung Cancer**												
A549/ATCC	2.44	6.24	45.2	2.06	4.64	12.9	2.09	4.64	17.5	2.83	8.97	38.8
EKVX	1.61	3.05	5.78	1.56	2.98	5.70	1.61	3.12	6.05	1.60	3.12	6.08
HOP-62	2.41	6.98	37.5	2.27	6.74	28.8	1.74	3.23	6.01	4.26	14.1	37.7
HOP-92	1.85	3.30	5.91	1.80	3.33	6.13	2.01	3.51	6.12	1.73	3.17	5.83
NCI-H226	1.93	4.52	14.1	1.78	4.24	16.2	1.84	4.52	>100	2.82	8.35	46.1
NCI-H23	1.47	3.30	7.45	1.44	3.55	8.71	1.35	2.85	6.03	1.53	3.15	6.49
NCI-H322M	2.50	7.69	27.6	2.22	7.91	29.7	1.68	3.74	8.29	3.42	12.8	35.8
NCI-H460	3.22	12	52.7	3.02	11.4	45.4	3.08	10.4	49.7	2.68	8.07	82.6
NCI-H522	1.81	3.36	6.24	1.74	3.28	6.18	1.75	3.22	5.94	1.56	3.06	6.02
**Colon Cancer**												
COLO 205	1.78	3.87	ND ^a^	1.73	3.66	7.74	1.86	3.64	7.12	1.68	3.08	5.63
HCC-2998	1.87	3.46	6.42	1.94	3.52	6.39	1.76	3.25	6.00	1.81	3.26	5.88
HCT-116	1.53	4.16	44.3	1.05	2.31	5.05	1.32	2.79	5.92	1.53	2.88	5.45
HCT-15	1.33	3.02	6.83	1.13	2.55	5.76	1.30	2.94	6.66	1.54	3.92	ND ^a^
HT29	2.13	4.49	9.46	3.40	15.2	99.5	2.87	7.08	62.5	1.87	3.48	6.48
KM12	1.49	3.08	6.40	1.41	2.89	5.92	1.44	3.07	6.51	13.0	26.6	54.7
SW-620	1.84	3.73	7.56	1.60	3.21	6.41	1.83	3.51	6.73	1.76	3.47	6.85
**CNS Cancer**												
SF-268	1.53	3.05	6.06	1.47	2.94	5.87	1.55	3.09	6.14	3.25	14.4	72.7
SF-295	1.85	3.42	6.32	2.03	4.20	8.70	1.86	3.43	6.30	1.99	5.41	20.8
SF-539	1.68	3.13	5.81	1.67	3.09	5.71	1.68	3.13	5.84	1.79	3.22	5.78
SNB-19	1.81	3.72	7.64	1.80	3.97	8.73	1.68	3.16	5.94	5.82	18.8	44.2
SNB-75	1.50	2.84	5.39	1.63	3.43	7.21	1.46	2.79	5.33	1.94	5.37	23.2
U251	1.85	3.73	7.53	1.70	3.27	6.28	1.80	3.54	6.94	1.92	4.44	10.8
**Melanoma**												
LOX IMVI	0.952	2.36	5.70	0.839	2.22	5.33	1.06	2.33	5.14	1.10	2.32	4.88
MALME-3M	1.71	3.19	5.95	1.36	2.77	5.64	1.65	3.16	6.02	1.76	3.28	6.11
M14	1.82	3.81	7.96	1.46	2.83	5.50	1.68	3.23	6.21	1.69	3.09	5.66
MDA-MB-435	1.70	3.11	5.67	0.266	0.671	2.45	0.442	1.53	3.99	1.72	3.57	7.39
SK-MEL-2	1.81	3.34	6.19	1.44	2.86	5.69	1.63	3.03	5.64	1.59	3.10	6.03
SK-MEL-28	1.85	3.34	6.02	1.71	3.14	5.79	1.73	3.14	5.70	1.79	3.23	5.85
SK-MEL-5	1.75	3.17	5.74	1.74	3.13	5.65	1.72	3.11	5.62	1.62	2.97	5.47
UACC-257	1.57	3.09	6.11	1.28	2.61	5.34	1.39	2.78	5.58	1.51	2.85	5.37
UACC-62	1.82	3.37	6.25	1.80	3.36	6.28	1.81	3.29	5.97	1.77	3.42	6.60
**Ovarian Cancer**												
IGROV1	1.46	2.92	5.86	1.37	2.82	5.84	1.36	2.76	5.61	1.73	3.55	7.26
OVCAR-3	1.12	2.40	5.15	0.759	2.03	4.54	0.962	2.19	4.87	ND ^a^	ND ^a^	ND ^a^
OVCAR-4	1.21	2.49	5.11	1.00	2.22	4.91	1.13	2.37	4.97	1.40	2.82	5.66
OVCAR-5	1.86	3.42	6.29	1.84	3.33	6.03	1.92	3.46	6.25	2.38	5.79	23.7
OVCAR-8	1.61	3.25	6.59	1.49	2.92	5.70	1.57	3.01	5.77	1.64	3.28	6.56
NCI/ADR-RES	2.03	5.16	34.6	1.89	5.09	>100	1.70	4.11	9.93	1.81	4.47	21.7
SK-OV-3	12.6	27.5	59.8	13.5	27.1	54.2	11.0	30.1	82.3	12.6	25.1	50.1
**Renal Cancer**												
786-0	1.91	3.61	6.82	1.80	3.25	5.86	1.72	3.19	5.89	1.52	2.88	5.44
A498	1.90	4.60	13.1	2.65	10.8	34.8	1.56	5.52	24.4	2.35	11.4	33.8
ACHN	1.47	2.80	5.34	1.35	2.69	5.35	1.57	2.92	5.45	1.67	3.04	5.51
CAKI-1	1.50	2.87	5.47	1.45	2.84	5.57	1.60	2.97	5.51	1.76	3.15	5.65
RXF 393	1.42	2.76	5.36	1.44	2.78	5.39	1.57	2.96	5.61	1.63	2.99	5.48
SN12C	1.70	3.29	6.36	1.59	3.04	5.80	1.58	2.99	5.66	1.71	3.25	6.19
TK-10	2.34	3.86	6.36	2.25	3.72	6.18	2.30	3.77	6.18	2.32	3.78	6.15
UO-31	1.42	2.73	5.26	1.40	2.71	5.23	1.43	2.75	5.28	1.69	3.06	5.53
**Prostate Cancer**												
PC-3	1.21	2.58	5.49	1.26	2.77	6.11	1.29	2.77	5.93	2.56	10.4	43.6
DU-145	1.72	3.11	5.62	1.63	2.99	5.49	1.72	3.13	5.68	1.84	3.55	6.83
**Breast Cancer**												
MCF7	1.78	4.49	>100	1.33	3.65	ND ^a^	1.54	3.82	9.44	1.82	5.33	>100
MDA-MB-231/ATCC	1.19	2.53	5.37	1.07	2.38	5.31	1.17	2.50	5.34	1.46	3.08	6.48
HS 578T	2.27	5.64	>100	2.11	5.25	>100	2.22	5.82	>100	2.43	5.78	>100
BT-549	1.72	3.27	6.19	1.52	2.88	5.46	1.43	2.80	5.49	1.97	3.40	5.87
T-47D	1.37	ND ^a^	>100	0.467	3.17	>100	1.01	3.91	>100	1.50	3.62	8.74
MDA-MB-468	1.52	3.06	6.16	1.45	2.88	5.71	1.49	2.92	5.75	1.23	2.57	5.37

^a^ ND means “not determined”.

**Table 3 biomedicines-12-01241-t003:** IC_50_ values of **AQQ1** and **AQQ2** were determined from dose-response curves of MTT assay. Values expressed as mean ± SEM.

(µM)		DU-145	MDA-MB-231	HCT-116	HUVEC
**AQQ1**	IC_50_	4.18 ± 0.73	8.27 ± 0.91	5.83 ± 0.76	5.39 ± 0.73
**AQQ2**	IC_50_	4.17 ± 0.80	13.33 ± 1.12	9.18 ± 0.96	6.39 ± 0.80
**DOXO**	IC_50_	<100	59.06 ± 9.06	24.35 ± 6.90	57.48 ± 9.82

**Table 4 biomedicines-12-01241-t004:** ADME profile of **AAQ1** and **AQQ2** (in vitro).

Source	Parameters	AQQ1	AQQ2	Verapamil	Atenolol
	Log P	1.85	1.77	−0.21	−0.33
	Log D	2.37	2.35	1.87	−1.77
Mouse liver microsomes	% Metabolism in 30 min	63	96.5	80.0	
	Half-life (min)	20.5	5.5	11.0	
	CL_int_ (μL/min/mg protein)	68.5	249	130.0	
Rat liver microsomes	% Metabolism in 30 min	100	100	79.0	
	Half-life (min)	2	3	11.0	
	CL_int_ (μL/min/mg protein)	655.5	501	121.0	
Dog liver microsomes	% Metabolism in 30 min	41.5	79.5	81.0	
	Half-life (min)	31	17	11.0	
	CL_int_ (μL/min/mg protein)	45	84	131.0	
Human liver microsomes	% Metabolism in 30 min	39.5	47	79.0	
	Half-life (min)	>30	26.5	11.0	
	CL_int_ (μL/min/mg protein)	39	52	129.0	

All the compounds partitioned towards *n*-octanol (lipophilic).

**Table 5 biomedicines-12-01241-t005:** Mean pharmacokinetic parameters of **AQQ1** and **AQQ2** following IV and PO administration to Sprague–Dawley rats.

Parameters	AQQ1		AQQ2	
	IV	PO	IV	PO
C_0_ (IV only)/C_max_ (PO only) (ng/mL)	90.8 ± 71.3	435 ± 18.1	23.2 ± 12.8	282 ± 79.4
T_max_ (h)	NA	0.5	NA	1
AUC_last_ (h × ng/mL)	77.5 ± 9.81	3810 ± 111	77 ± 21	2300 ± 416
AUC_INF_ (h × ng/mL)	115 ± 45.7	4000 ± 179	110 ± 18.1	2370 ± 375
T_1/2_ (h)	9.35 ± 8.75	5.22 ± 0.843	12.8 ± 2.95	4.72 ± 1.16
V_d_ (L/kg)	90.3 ± 40	NC ^b^	147 ± 9.71	NC ^b^
Cl (mL/min/kg)	161 ± 62.9	NC ^b^	154 ± 26.9	NC ^b^
MRTlast (h)	3.77 ± 0.279	7.47 ± 0.367	6.84 ± 1.84	6.61 ± 0.787
F%	NA ^a^	9.83	NA^a^	5.96

^a^ NA means “not available”; ^b^ NC means “not calculate”.

## Data Availability

The datasets used and/or analyzed during this study are available from the corresponding author on reasonable request.

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
