# Peer review of "Prospects for Prostate Cancer Chemotherapy: Cytotoxic Evaluation and Mechanistic Insights of Quinolinequinones with ADME/PK Profile"

_biomedicines, 2024, doi:10.3390/biomedicines12061241_

Round 1

Reviewer 1 Report

Comments and Suggestions for Authors

Paper titled (Prospects for Prostate Cancer Chemotherapy: Cytotoxic Evaluation and Mechanistic Insights of Quinolinequinones with ADME/PK Profile) by  Jannuzzi et al. discussed the the cytotoxic effect of Quinolinequinones on prostate cancer cells and pharmacokinetic profile. 

1-Title: should be revised to be more informative without overestimation, Start directly with what was the action of the Quinolinequinones on prostate cancer and mention if in vitro or in vivo & mechanism how these actions were mediated

Why prostate cancer only is mentioned in the title??

2- Abstract should be amended by some numerical values from results& follow the journal guidelines for word count

3- Key words : why included breast cancer? try to revise them to give the best indexing key words

4- Introduction is not concrete & talks first about large intestine & colon, then breast cancer & then prostate cancer. It is assumed (prostate cancer) is the most important topic and mentioned in the title, please revise the whole INTRO to be focusing on the topic & mention the novelty, aim and how you achieved it

5- Plagiairism % is high and must be reduced to less than 15%

Single sources are copied for 12% and 6%, please consider this seriously

6- - What was the age and weight of miceat the begin of the experiment?

7- Ethical approval number & date must be provided

8- Ensure every abbreviation is explained at the first appearnace in abstract & then in the body text
9- Animal details and housing should be separate from the experiment design. Give exact number of groups and names in details or a table
10- Methods in general lacks references at many occasions.
11- - Authors should give the source of chemicals, kits and antibodies completely and consistently (code, company, town, state and country) & version for software
12- Every abbreviation in figures should be explained in the figure legend to be self explanatory & stands alone.
13- Authors should confirm in methods that "every possible comparison between the study groups was considered" and apply this in results.
14- Authors have to check the normality of distribution of the results by a suitable post hoc test (such as Shapiro-Wilk test or K-S test) before deciding to choose certain ANOVA. If the normality test indicated normal dist of the data, so use one-way ANOVA, if not, use non parametric ANOVA. In all cases choose a suitable post-hoc test
15- Use appropriate abbreviations for minutes, seconds...etc
16- Mention "n" in each illustation individually
17- No need to mention P <0.001 or P<0.01; as P<0.05 is enough and high p levels does not mean the mean values are greatly different; just tell that the SD values are small.   Q 2   Check List  Reviewer 1: Sawsan A Zaitone | 28 Mar 2024 | 11:56 #1 a. Is the quality of the figures and tables satisfactory?
- No

b. Does the reference list cover the relevant literature adequately and in

Comments on the Quality of English Language

fine

Author Response

Dear reviewer,

Please see the attached document for your consideration.

Reviewer 2 Report

Comments and Suggestions for Authors

Manuscript biomedicines-2931278

Prospects for Prostate Cancer Chemotherapy: Cytotoxic Evaluation and Mechanistic Insights of Quinolinequinones with ADME/PK Profile” for Biomedicines

Comments:

1. In the description of table 1, please complete what the obtained results indicate? The current description is not clear. Is this a decrease in viability compared to the control?

2. Figure 4. Please insert a scale bar on the images. In addition, please provide the results of scratch measurements compared to the control in the description.

3. Figure 6. The control result indicates very poor viability of the cell culture. I would like to ask the authors whether such a low cell viability does not affect the entire experiment, in terms of the sensitivity of the cells to the applied compound?

4. Experimental. 3.1.1. What concentration of DMSO was at a sample compound concentration of 10 microM? Did the solvent concentration affect the viability of the cell culture?

Author Response

(The authors gave the same response as above.)

Reviewer 3 Report

Comments and Suggestions for Authors

On request of Biomedicines, I have revised the manuscript titled “Prospects for Prostate Cancer Chemotherapy: Cytotoxic Evaluation and Mechanistic Insights of Quinolinequinones with ADME/PK Profile” by Ayse Tarbin Jannuzzi and colleagues.

The main scope of this study was to detect new possible molecules to treat prostate cancer, among five quinolinequinones (AQQ1-5) that have previously been described against the whole panel of National Cancer Institute (NCI) cancer cell lines based on the NCI Developmental Therapeutics Program. To this end, the authors have carried out a series of in vitro biological assessments and in vivo pharmacokinetic studies to gain a deeper understanding about AQQ1 and AQQ2 compounds, finding AQQ1, as the compound with higher activity and better PK parameters in rat. In silico studies identified MAPK as one of the probable targets for AQQ1.

COMMENTS

Together with breast and colon cancer, prostate cancer, affecting mostly men over 50 years of age, causes millions of deaths worldwide each year. Consequently, extensive research which could lead to the discovery of new chemotherapeutic template molecules for prostate cancer treatment is relevant. In this regard, the topic proposed by authors could be of high interest for experts in the field, but my major concern is about its lack of novelty. The present work is a mere reproduction of a previous work by the authors on the same class of compounds (https://doi.org/10.1002/cbdv.202300848). In the previous work, compounds AQQ1-AQQ5 and then specificallyAQQ2 and AQQ3 were essayed using experiments and cell lines identical to those used here to re-essay compounds AQQ1-AQQ5 (curiously with significantly different results) and then AQQ1 and AQQ2. Authors should add further different experiments to better differentiate their products. Additionally, except for Sections 3.1.4 and 3.1.5, all other Section 3.1.1-3.1.3, 3.1.6-3.1.8, 3.2.1-3.2.3 and 3.3 should be replaced by simple citation of https://doi.org/10.1002/cbdv.202300848.

Other minor issues

Please, check all manuscript, and where missing, add specifications for the abbreviations at their first mention in each Section.

Figure 1 is the same previously reported (https://doi.org/10.1002/cbdv.202300848) and should be moved in Supplementary Materials.

While do results in Table 1 in the present manuscript strongly differ from those reported in Table 1 in https://doi.org/10.1002/cbdv.202300848, as well as those results in Table 2 strongly differ from those reported in Tables 2-4 in https://doi.org/10.1002/cbdv.202300848, when tests were carried out with the same compound on the same cell line?

Table 1 title. Please, standardize the size of characters at the values indicated by the template.

The way to provide the references in the main text in not correct. Please, revise.

Figure 4 caption. Please, correct DU145 with DU-145.

Figure 6. Please, change the colour (black) in the bars graph for life cell, so that the error bars could be visible.

While do results of cytotoxicity of AQQ2 in Figure 3 and Table 3 in the present manuscript strongly differ from those reported in Figure 3 and Table 5 in https://doi.org/10.1002/cbdv.202300848, when tests were carried out with the same compound on the same cell line?

Line 142. Please, change “&” with “and”.

On these considerations, this study supplies interesting but not relevant information, if compared to that previously reported by the same authors. In addition the contents of this manuscript should be not only enriched but also differently organized and presented, compared to the previous work. I ask authors to address all the above-mentioned issues for making the present paper worthy of further consideration for publication on Biomedicines.

Comments on the Quality of English Language

Minor editing of English language required

Author Response

(The authors gave the same response as above.)

Round 2

Reviewer 1 Report

Comments and Suggestions for Authors

The revised version of paper titled (Prospects for Prostate Cancer Chemotherapy: Cytotoxic Evaluation and Mechanistic Insights of Quinolinequinones with ADME/PK Profile) by Ayse Tarbin Jannuzzi et al. was improved to some extent but there are still spme problems that must be resolved first before giving a recommendation.

1- The introduction is too long and must be shortened to be more concrete and focusing on the topic. Need to explore the novely & the aim of the study well.

2- Plagiraism report shows high % of plagiairism which is NOT accepted, Authors must solve this by rewritteing NOT just claiming that this is a self plagiarism or so on as mentioned in the reply. Without solving this issue, the paper must be rejected

One sources shows 12% and 7% also NOT accepted and No single source should show more than 3%

total plagiairsm should not be more than 15%

Comments on the Quality of English Language

moderate

Reviewer 2 Report

Comments and Suggestions for Authors

Thank you Authors for providing corrections. The manuscript is in current form sufficiently improved. 

Reviewer 3 Report

Comments and Suggestions for Authors

Dear Authors,

thank you for your explanation and work of revision.

Round 3

Reviewer 1 Report

Comments and Suggestions for Authors

thanks for doing the corrections

Comments on the Quality of English Language

FIne